# Epidemiology of sports-related fatalities during organized school sports in Japanese high schools between 2009 and 2018

**Miwako Suzuki Yamanaka**[1]◉, **Yuri Hosokawa**[2‡], **Mamoru Ayusawa**[3‡], **Norikazu Hirose**[2‡], **Koji Kaneoka**[2]◉*

**1** Graduate School of Sport Sciences, Waseda University, Tokorozawa, Saitama, Japan, **2** Faculty of Sport Sciences, Waseda University, Tokorozawa, Saitama, Japan, **3** Department of Pediatrics and Child Health, Nihon University School of Medicine, Itabashi-ku, Tokyo, Japan

◉ These authors contributed equally to this work.
‡ These authors also contributed equally to this work.
* kaneoka@waseda.ne.jp

## Abstract

Limited literature has investigated epidemiology of sports-related fatalities during high school organizes sports in Japan. Therefore, the purposes of this study are to determine the frequency and incidence rate of sports-related fatalities in Japanese high schools by cause and sports, and to examine the type of on-site first responder. Insurance claim data of sports-related fatalities in Japanese high schools reported to Japan Sports Council Injury and Accident Mutual Aid Benefit System between 2009 and 2018 were retrieved as the primary data source. All fatalities were classified into direct or indirect type by the reported etiology and further categorized into cardiac-related, head and neck injury, exertional heat stroke (EHS), or other. Frequency and incidence rate were calculated by cause of death and sports, and incidence rates were expressed per 100,000 athlete-years (AY) with 95% confidence interval (CI). Information regarding first responder to the incident was also retrieved and examined by frequency. A total of 63 sports-related fatalities were analyzed. The overall incidence rate was 0.45 (95%CI = 0.25–0.65) per 100,000AY. The incidence rates of direct and indirect fatalities declined from 0.36 and 0.50 per 100,000AY to 0.28 and 0.00 per 100,000AY, respectively. The leading cause of deaths was cardiac-related (n = 30/63, 47.6%), followed by head and neck injury (n = 15/63, 23.8%) and EHS (n = 14/63, 22.2%). The number of fatalities was highest in male baseball (n = 12/63, 19.0%) and the incidence rate was highest in male judo (4.79 per 100,000 AY, 95%CI: 0.68–8.15). Coach was the most frequently reported first responder onsite (n = 52/63, 82.5%). Medically trained personnel were involved in onsite care in two cases (3.2%). In conclusion, the occurrence of sports-related fatalities has declined over time from 2009 to 2018. To deliver appropriate medical care onsite for better survival, employment of medically trained personnel should be promoted in high school sports setting in Japan.

## Introduction

Sports participation provides a wide variety of benefits to student athletes, such as physical and psychological well-being [1, 2], and improved academic performance [3]. However, injury

**Data Availability Statement:** The data underlying the results presented in the study are available from Japan Sports Council at https://www.

jpnsport.go.jp/anzen/anzen_school/anzen_school/
tabid/822/Default.aspx.

**Funding:** The author(s) received no specific funding for this work.

**Competing interests:** The authors have declared that no competing interests exist.

occurrence paralleled with sports participation has been an issue that cannot be overlooked [4]. Especially when catastrophic injuries or deaths occur during school organized sports, these tragedies gather massive attention and significantly impact the community. Thus, the health and safety of student-athletes should always be one of the top priorities in school organized sports. Sports-related injury and illness prevention is particularly important at the high school level in Japan because the frequency of sports-related fatality has been higher in high schools than in elementary and middle schools according to the number of insurance claims reported to Japan Sports Council (JSC) Injury and Accident Mutual Aid Benefit System. Despite the fact that sports-related fatalities occur every year during school-sanctioned sports activities in Japan, resources to implement proactive safety strategies are yet to be set in place. For example, access to athletic trainers (ATs) are very limited in organized school sports in Japan, whereas 66–70% of high schools in the United States (US) have access to ATs who play important roles in implementing safety management for their student-athletes as an allied medical professional [5–8]. Additionally, coaches in Japanese high schools have limited opportunities to obtain training on first aid management of sports related medical conditions. This is partially due to a lack of standardized requirement for such training for high school coaches in Japan. In addition, most of these coaches are faculty members of the school, who have been assigned to coach and supervise respective sports regardless of their familiarity with the sport. Therefore, the chance of these faculty-member coaches to voluntary participate in first aid training in their busy schedule is limited. For example, a survey conducted by Japan Sports Agency in 2017 revealed that 71.4% and 66.2% of sports coaches who work at public and private high schools, respectively, had no sports instructor qualification set forth by Japan Sports Association (JASA) or national governing body of their assigned sports. In 2019, JASA introduced a new curriculum for the sports instructor qualification program, which required attendees to take 24 hours of educational course on sports medicine including basic life support (BSL) training. However, the new program merely ensures a one-time completion of basic life support (BLS) training and does not guarantee current knowledge and skills. Consequently, even with the recent updates in instructor qualification program, it is reasonable to assume the level of first aid care provided at the time of serious injury during sports by coaches in Japan is limited because (1) completion of training is voluntary and (2) there is a lack of follow-up requirement to receive BLS training after the initial completion. These circumstances may contribute to an increased risk of more serious injuries that may lead to morbidity and mortality [9].

To ensure the health and safety of student-athletes, prevention strategies need to be developed, and detailed epidemiological data is a fundamental element for evidence-based injury prevention [10]. However, limited literature has investigated epidemiology of fatalities specific to Japanese high school sports setting [11, 12]. Therefore, the main purpose of the current study is to determine the frequency and incidence rate of sports-related fatalities that happened during organized high school sports in Japan by their cause and sports, using insurance data reported to JSC. Our secondary aim was to retrospectively examine fatal incidents reported to JSC that occurred during organized high school sports in Japan to describe initial care provided at the time of incidents.

## Methods

### Data source

Principal investigator (MSY) acquired insurance claims data of fatal incidents that occurred during organized school sports at Japanese high schools from 2009 to 2018 from JSC. JSC insurance claims data were used as the primary data source since most high school students in

Japan (97.7% in 2018 academic year) are under the coverage of JSC Injury and Accident Mutual Aid Benefit System, which provide insurance coverage for permanent disabilities and fatalities that occur during school activities. For the purpose of this study, we examined only the cases that occurred during school organized sports. The case information received from JSC included sex and cause of death of the victim, location and time of the incident, sports and activity classifications by type (e.g., practice, conditioning, and game) and frequency (e.g., single-session, double-session, and training camp), and post-event onsite management. The onsite responder who provided initial care to the victim was identified from the description of onsite management found in the case reports. Training camp was classified under frequency because it is characterized by a successive day of multiple-session practices. Brief summary of each incident was retrieved from the database publicly available on the JSC website (https://www.jpnsport.go.jp/anzen/anzen_school/anzen_school/tabid/822/Default.aspx). Case information retrieved directly from JSC and the website were matched based on the pre-designated identification number by JSC.

Participation data of all sports except baseball was retrieved from All Japan High School Athletic Federation website (https://www.zen-koutairen.com/f_regist.html). Since baseball is independently governed by Japan High School Baseball Federation, participation data for baseball was retrieved separately (http://www.jhbf.or.jp/data/statistical/index_koushiki.html). Weather data (ambient temperature and humidity) was obtained from the database publicly available on the Japan Metrological Agency website (https://www.data.jma.go.jp/obd/stats/etrn/index.php) for heat-related. The data from the nearest meteorological station and within one hour of the incident was used as the representative data point. This study was exempted from institutional review board approval. This study was exempted from ethical review because it only involves the use of existing collections of data. Additionally, the data only contains non-identifiable information about human beings.

## Injury definition

Sports-related fatalit*y* include injuries which resulted directly from participation in the fundamental skills of the sport (direct injuries), or injuries which are caused by systemic failure as a result of exertion while participating in an activity or by a complication which was secondary to a non-fatal injury (indirect injuries), as defined by the National Center for Catastrophic Sport Injury Research (NCCSIR) (https://nccsir.unc.edu/definition-of-injury/). Type of the fatality (direct vs. indirect) was determined based on the cause of death included in the insurance claim report and the descriptive case summary of the incident retrieved from online database available on the JSC website.

Specific cause of death included in the insurance claim report provided by JSC were further categorized into four groups: cardiac-related, head and neck, exertional heat stroke (EHS), and other. Upon categorization of cause of death, experienced researchers whose specialties are in cardiovascular pathologies (MA), spine (KK), and heat-related illness (YH) were consulted. When the cause of death was unknown and no specific etiology was identified from the case summary, the case was initially labeled as probable cardiac-related death and reassessed by the researcher (MA) to determine the final decision to include or exclude as cardiac-related case.

## Data analysis

Frequency and incidence rate per 100,000 athlete-years (AY) were calculated by year and type of fatality (direct vs. indirect), using the number of participants in the respective year in the rate denominator. Overall sports-specific incidence rates between 2009 and 2018 were determined by using total participants in the sports during the study period in the rate denominator

and expressed per 100,000AY with 95% of confidence intervals (CI) for sports in which five or more fatal events occurred during the study period. Descriptive statistics, such as frequency and proportion, were calculated for post-event initial care.

## Results

A total of 114 sets of case information were provided by JSC. Of the 114 cases, 51 were excluded before the analyses due to one of the following reasons: 1) occurred outside organized school sports (e.g., physical education class and school-hosted sports event) or 2) were not sports-related. The cases that were determined as non-sports-related included but not limited to fatalities due to lightning, avalanche, and traffic accident.

### Cause of fatality

Type and cause of fatality is summarized in Fig 1. Of 63 sports-related fatalities that occurred due to organized high school sports in Japan, 74.6% (n = 47) were indirect and 25.4% (n = 16) were direct injuries. The majority of indirect fatalities (n = 30/47) and approximately half of all fatalities (47.6%, n = 30/63) were cardiac-related. Of 30 incidents categorized into cardiac-related, 20 were confirmed of cardiac origin and ten were of unknown cause but judged highly likely to be cardiac-related based on case information by researcher (MA). Head and neck injuries (23.8%, n = 15) were the second most frequent cause of fatalities, and EHS was the third (22.2%, n = 14). Other accounted for 6% (n = 4) and included two non-traumatic sub-arachnoid hematoma, one traumatic abdominal aortic rupture, and one hydrocephalus.

### Incidence rates by type of fatality and year

The incidence rate of indirect and direct fatalities by academic year are summarized in Fig 2. The incidence rates of direct and indirect fatalities have declined from 2009 to 2018. The overall incidence rate of sports-related fatality through 2009 to 2018 were 0.45 per 100,000 athlete-years (95% CI = 0.25–0.65).

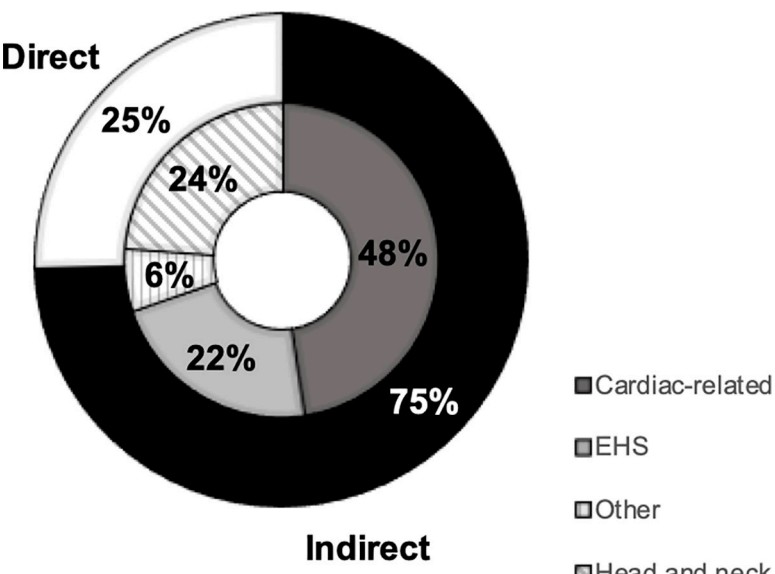

**Fig 1. Type and cause of sports-related fatalities reported to Japan Sports Council that occurred during organized high school sports between 2009 and 2018 in Japan.** Abbreviation: EHS, exertional heat stroke.

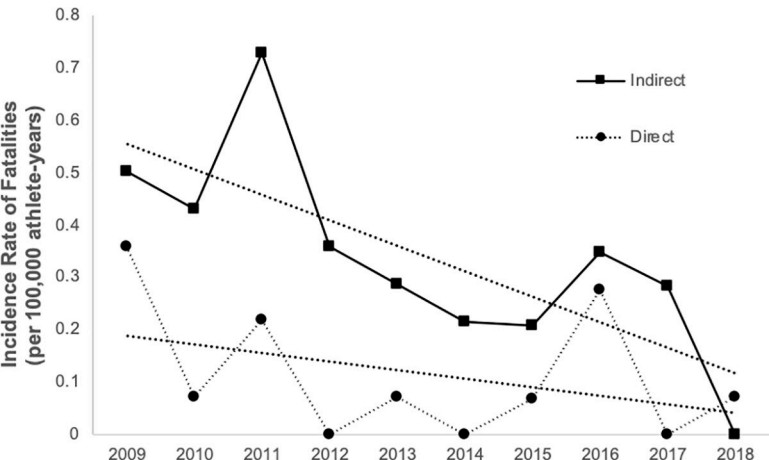

**Fig 2. Incidence rate of direct and indirect fatalities reported to Japan Sports Council that occurred during high school organized sports between 2009 and 2018 in Japan.**

## Fatalities by sports

Frequency of sports-related fatalities in male and female combined by sports is summarized in Fig 3. Baseball was the highest in frequency (n = 12), accounting for 19% of total fatalities. Direct fatalities (head and neck category) were the major cause only among judo and rugby. The overall incidence rates in male sports is shown in Table 1. Judo was the highest in incidence rate (4.41 per 100,000 athlete-years, 95% CI = 8.15–0.68), and rugby was the second highest (2.88 per 100,000 athlete-years, 95% CI = 4.84–0.91). Incidence rates in female sports were not calculated because the number of incidents were too few to determine accurate incidence rates.

## Onsite responder

The type of personnel who served as the first responder at the time of incident is summarized in Fig 4. Coach was involved in the post-event immediate care in most of the cases (82.5%,

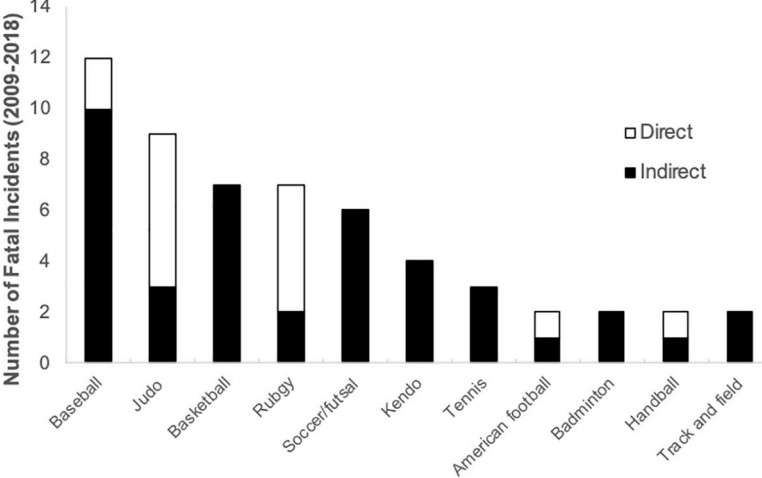

**Fig 3. Number of fatalities reported to Japan Sport Council during organized high school sports between 2009 and 2018 in Japan by different sports and cause (direct vs. indirect).**

**Table 1. Overall incidence rate of sports-related fatalities in male athletes reported to Japan Sports Council that occurred during high school organized sports between 2009 and 2018 in Japan.**

| Sports | Overall incidence rate per 100,000 athlete-years | 95% CI | Number of fatalities | Number of athletes |
|---|---|---|---|---|
| Judo | 4.79 | 0.68–8.15 | 9 | 187,787 |
| Rugby | 2.91 | 0.91–4.84 | 7 | 240,604 |
| Baseball | 0.68 | 0.15–1.19 | 12 | 1,765,428 |
| Basketball | 0.64 | 0.00–1.52 | 6 | 931.171 |
| Soccer/futsal | 0.38 | 0.05–0.72 | 6 | 1,592,876 |

Abbreviation: CI, confidence interval.

n = 52). Student was involved in 44.4% (n = 28), and school nurse teacher was involved in 15.9% (n = 10) of the cases. Medically trained personnel (physician or [athletic] trainer) served as the first responder in only two (n = 2/63) cases.

## Exertional heat stroke fatalities

A total of 14 heat-related incidents were reported. Representative ambient temperature and humidity data were available in 14 cases (average, 30.4 ± 2.71˚C) and 10 cases (average, 57 ± 7.57%), respectively. Information regarding the height and weight was available for 8 EHS victims (n = 8/14). The average body mass index (BMI) of these victims was 29.5 ± 7.3. The number of fatalities by month and activity frequency is illustrated in Fig 5. EHS fatalities occurred most frequently in August (n = 6) and followed by July (n = 5). Three EHS fatalities occurred on double-sessions days in July and August, respectively. In August, two additional EHS fatalities occurred at training camp.

Immediate onsite care provided at the time of incident is summarized in Table 2. Body temperature was measured in 35.7% (n = 5/14); however, the method of temperature measuring was unknown in all cases. In 64.3% (n = 9/14) of the total cases, onsite cooling was initiated. The most frequently used cooling method was application of ice bags (n = 8). Onsite cooling was not initiated in 35.7% (n = 5/14).

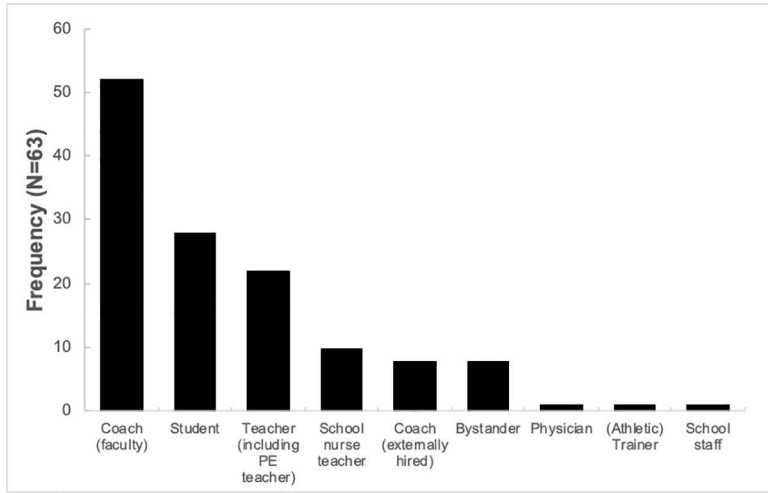

**Fig 4. Distribution of first responder by type reported in 63 confirmed sports-related catastrophic events reported to Japan Sports Council.**

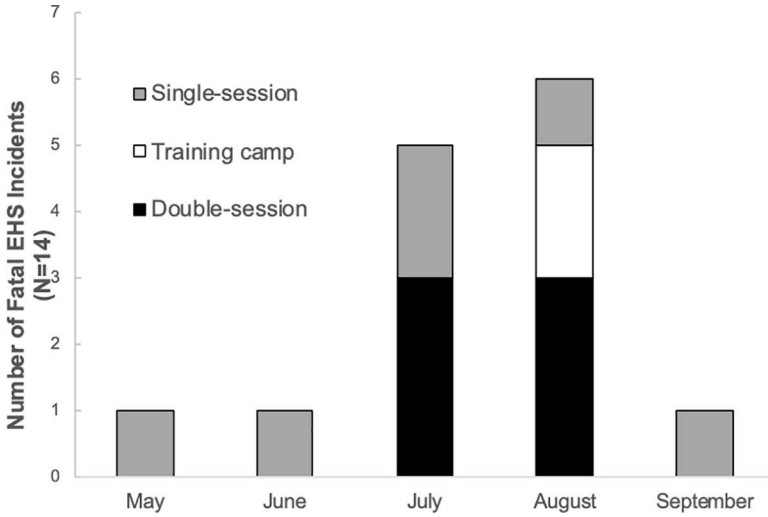

**Fig 5. Number of fatal EHS (exertional heat stroke) incidents reported to Japan Sport Council that occurred during organized high school sports in Japan between 2009 and 2018 by month and activity frequency.**

## Discussion

There is limited literature that describes the frequency and incidence rates of sports-related fatalities that happened during school organized sports in Japanese high schools by cases and different sports [13]. Although the frequency and incidence of indirect and direct fatalities both have decreased through 2009 to 2018, the extent of reduction was greater in indirect fatalities. Reduction in both of cardiac-related and EHS fatalities contributed to the greater reduction in indirect fatalities. In Japan, AEDs became publicly available in 2004 [14]. Since then, continued effort to distribute AEDs to schools nationwide was made by multiple organizations, such as Japan Circulation Society and Japanese Society for Emergency Medicine, and in 2009, 98% of high schools in Japan have at least one AED installed on campus, according to a survey by the Ministry of Education, Culture, Sorts, Science and Technology (MECSST). In 2014, MECSST made the first recommendation to provide BSL training to school faculty. In the following year, the AED committee of the Japanese Circulation Society affirmed the

**Table 2. Temperature assessment and immediate onsite care for 14 fatal EHS (exertional heat stroke) reported to Japan Sports Council that happened during high school organized sports in Japan between 2009 and 2018.**

| **Temperature measured** | | |
|---|---|---|
| | Yes | 5 |
| | No | 2 |
| | Unknown | 7 |
| **Onsite cooling initiated** | | |
| | **Yes** | **9** |
| | Ice bag application | 8 |
| | Fanning | 5 |
| | Moving to shaded area | 5 |
| | Dousing with water | 4 |
| | Moving into air-conditioned area | 2 |
| | Wet towel application | 2 |
| | Cold water immersion | 0 |
| | **No** | **5** |

necessity to introduce BLS training program to schools [15]. As a result, MECSST reported that 75.6% of high schools in Japan provided BLS training to their students, and 58.0% of high schools provided the training to all faculty members by March 2014 [14]. Kiyohara *et al.* [16] reported that the proportion of sudden cardiac arrest (SCA) cases with immediate AED application during school activities increased from 61.9% in 2008 to 87.0% in 2015. Similarly, the survival rate with favorable neurological outcome post 30 days of SCA improved from 38.1% in 2008 to 56.5% in 2015 [16]. Whilst the data from Kiyohara *et al.* [16] is not specific to sports-related activities, their findings may be associated with the reduction in indirect fatalities observed in this study. Since 2007, JSC has been addressing prevention of sports-related injury and illness by analyzing more than one million insurance claim data reported every year to the Injury and Accident Mutual Aid Benefit System and providing resources for schools to help promote school sports safety. For example, JSC issued their investigation report on prevention of sudden death during school activities in 2010 and head and neck trauma during school sports in 2012. Their unique position that almost all schools in Japan subscribe to their insurance policy provides them the advantageous opportunities to disseminate their findings. The advocation by JSC may have contributed to the lower number of sports-related fatality occurrence in recent years in Japanese high schools.

Although the strategy to distribute AED and promote BLS training to the teaching staff and students was successful to a certain degree, there is yet an area for further improvement. The results confirmed that coaches will most likely serve as the first responder; however, coaches in Japanese high schools have bare minimum requirement in BLS. Although schools might offer the opportunity to complete BLS training to their faculty members, 42% of faculty members in Japanese high schools still remained untrained by March 2014 as the survey by MECSST revealed [14]. Therefore, further advocacy of BLS training to coaches is necessary. Students are also frequently present and take immediate actions with coaches onsite during a catastrophic event (Fig 4). Thus, the effort to promote the BLS training to students, especially who participate in school sports, should be continued. The other major issue regarding immediate onsite care in Japanese high schools is the lack of medically trained personnel. Within the Japanese school system, school nurse teacher is often the only licensed professional who is trained to handle medical emergencies. However, the result of this study revealed that school nurse teachers were present only in 15.9% of the fatal event. This is likely because the work hours of school nurse end by after school when most school organized sports are scheduled. Physician or (athletic) trainer was present only in two cases (3.2%). This is remarkably low compared to the US, where 66–70% of high schools have access to ATs who are responsible for prevention and management of sports-related injuries and illness [5–8]. Causes of death revealed in our study (e.g., cardiac-related, head and neck, EHS) require appropriate prehospital management (i.e., rapid CPR, AED application, in-line stabilization of cervical spine, vigorous body cooling) to save lives and optimize patient outcome [16–18]. Thus, to ensure the health and safety of student-athletes, distribution of medically trained personnel to schools should be considered in Japan as well.

## Fatalities by sports

In the current study, male judo and rugby yielded the high incidence rates (4.79 and 2.91). Those two sports are unique in that the majority of the fatalities were of direct causes. This is likely due to the nature of the sports, which are characterized with frequent collisions. Previous study shows that 70% of severe judo head injuries occurred when the victim was thrown to the mat and that 63% of severe judo neck injuries occurred when executing offensive maneuver [19]. Direct injuries are partially preventable by regulating a high-risk technique and

appropriate coaching that considers safe execution of skills of the sports. In 2011, the All Japan Judo Federation implemented instructor qualification system that requires judo coaches to learn prevention and immediate management of catastrophic head and neck injuries [19]. Murata *et al*. [20] concluded that this system and the associated educational efforts were effective in preventing catastrophic head trauma in judo, which resulted in reduction of catastrophic head injury risk by 66% in middle and high school judo athletes in Japan.

Other characteristics that is unique in Japanese high school organized sports is the high frequency of baseball fatalities. The number of baseball fatalities was the highest among all sports (n = 12/63, 19.0%). This may be partially due to the large participation population of baseball in Japan. However, the high frequency of baseball fatalities in Japanese high school cannot be explained merely by the large participation since baseball is considered a low fatality risk sports in spite of the large population in the US [21]. In this study, the proportion of indirect fatalities accounted for 83.3% of all baseball related fatalities, whereas NCCSIR reported indirect fatalities accounted for 59.1% of all baseball related fatalities reported between 1982–1983 to 2017–2018 academic years [22]. On the other hand, in the US, football is regarded as the sports that has an increased risk of indirect fatality; the proportion of indirect fatalities in college football accounted for approximately 80% of all football related fatalities at the level, similarly to Japanese high school baseball [22]. Thus, as pointed out in college football in the US [23], the inherent design flaws of conditioning program that are characterized with excessive intensity and practice schedule that does not allow optimal recovery might have led to the high occurrence of baseball fatality among Japanese high schools. However, further investigation to examine the relationship between exercise intensity and occurrence of indirect catastrophic baseball injuries in Japanese high schools is warranted to determine the factors contributing to the high incidence of indirect fatalities in Japanese high school baseball.

## Exertional heat stroke

A total of 14 fatal EHS incidents were reported in our study period and they occurred between May and September. In Japan, summer break typically starts in mid to late July and ends in late August, providing opportunities for teams to schedule double-session practices and training camp during this period. The results from this study clearly showed that double-sessions and training camp add up additional fatal EHS incidents (Fig 5). Previous investigations suggest that the cumulative effect of heat exposure increases the risk of exertional heat illness [24, 25]. Thus, it is reasonable to assume that double-sessions and training camp that consists of consecutive days of multiple session practices in summer months are associated with the increased occurrence of EHS morbidity. In this study, the mean BMI of EHS victims was $29.5 \pm 7.3 \text{ kg/m}^2$, which is well beyond the healthy range (18.5–24.9 kg/m$^2$) [26]. This is consistent with previous studies that concluded increased BMI is a risk factor of exertional heat illness as those with greater BMI have limited ability to dissipate heat [27].

When EHS occurs, there are two key factors for survival; accurate internal body temperature measuring and adequate onsite cooling. The chance for survival and the extent of organ damage depend on the duration of excess internal body temperature above 40.5°C [17, 27]. Internal body temperature should be measured rectally to accurately diagnose EHS and determine the end point of cooling [17, 27, 28]. In this study, body temperature was measured only in 35.7% (n = 5/14) of the reported EHS cases. Although the method of temperature measuring was unknown in all cases, it is anticipated that axillary or tympanic thermometry was utilized since those are the common methods readily available for first responders in Japan [28]. The limitation of axillary and tympanic thermometry to assess body temperature of exercising individuals is well described in previous studies [28, 29]. Axillary thermometry gives a lower

estimate of body temperature, leading to underestimation of hyperthermia [28, 29]. Tympanic thermometry is vulnerable for invalid measures due to the difficulty in correctly measuring [28–30]. Conclusively, the post collapse assessment reported in this study falls short of evidence-based best practice in that the use of rectal thermometry was not reported in any cases. Moreover, the mode of onsite cooling selected were not appropriate in all reported cases. The gold standard of cooling method for EHS is cold water immersion (CWI) [17, 27, 28], and the survival rate when it was achieved to cool the internal body temperature to less than 38.9°C within 30 minutes of collapse is 100% [27]. Other methods, such as ice bag application that were most frequently used in the reported cases, are not able to achieve the adequate cooling rate for EHS [17]. The reason why CWI was not selected in any fatal EHS cases is likely the lack of awareness. In Japan, ice bag application to the major arteries has been traditionally considered as the first priority [28]. In fact, it is still explained as an acceptable method of cooling when ice tub and medically trained personnel to carry out CWI are absent in the domestic guidelines. In 2019, CWI was first introduced as one of recommended methods for EHS treatment in a guideline released by JASA. However, since medically trained personnel are rarely present in Japanese school sports setting, dousing with water and ice towel application are recommended as reasonable choices by JASA and JSC. The effectiveness of the shift away from ice bag application to newly recommended methods of cooling in preventing EHS fatality should be examined in the future.

## Limitations

This study is not without limitations. The primary data source of this study was insurance claims reported to the JSC Injury and Accident Mutual Aid Benefit System. Therefore, there may be undetected cases of fatality incidents in high school sports that was not reported to the system. As a nature of insurance data, there could be a gap between the year of incident occurrence and the year in which the data was filed in the database. Thus, there may be additional cases from our study period (2009–2018) that will become available for further analysis in the future. Moreover, since the database used in this study was not designed for a research purpose, key information regarding acute management (e.g., methods of body temperature assessment and implementation of emergency action plans) was not readily available. Development of a national surveillance system specific to sports-related catastrophic injuries in future may enable researchers to collect precise epidemiological data for analyses. Lastly, the sports-specific incidence rates should be interpreted with caution due to small case numbers. This specific limitation could be addressed through a future study that has longer study duration.

## Conclusion

In Japan, a total of 63 sports-related fatalities were reported at high school level between 2009 and 2018 academic years. Of the 63 cases, 74.6% (n = 47) were of indirect type, and 47.6% (n = 30) were cardiac-related death. Head and neck injuries (23.8%, n = 15) were the second most frequent cause of fatalities, followed by exertional heat stroke (EHS) (22.2%, n = 14). The incidence rate of indirect fatalities had remarkably declined from 2009 to 2018 over that of direct fatalities. The rapid expansion of AED installation in high schools may explain the reduction over time. Looking by sports, male judo and rugby yielded high incidence rates (4.79 and 2.91 per 100,000 athlete-years, respectively). This is likely due to the nature of the sports, which are characterized with frequent collisions as the majority of the fatalities in the two sports were of direct causes. Male baseball caused the greatest number of fatalities (n = 12) in the study period. The factors contributing to the high frequency of fatality in baseball in Japan should be examined in the future study. There were 14 fatal EHS incidents during the

study period. Of the 14, 11 (78.6%) occurred in July and August, and the majority occurred in double-sessions or training camp. In none of the EHS cases, CWI was utilized as a means of onsite cooling. Only two cases reported of having medically trained personnel at the time of event, and coaches were the most frequently reported first responder onsite (82.5%). Distribution of medically trained personnel in high school sports and improved coaching education to appropriately respond to medical emergency may further reduce morbidity and mortality during high school sports in Japan.

## Acknowledgments

The current study was a part of a joint research project between Japan Sports Council (JSC) and Waseda University. We would like to express our deepest gratitude to all staff members at JSC who have put tremendous efforts into this project.

## Author Contributions

**Conceptualization:** Miwako Suzuki Yamanaka, Norikazu Hirose, Koji Kaneoka.

**Data curation:** Miwako Suzuki Yamanaka, Koji Kaneoka.

**Formal analysis:** Miwako Suzuki Yamanaka, Mamoru Ayusawa.

**Investigation:** Miwako Suzuki Yamanaka, Mamoru Ayusawa, Koji Kaneoka.

**Methodology:** Miwako Suzuki Yamanaka, Yuri Hosokawa, Norikazu Hirose, Koji Kaneoka.

**Project administration:** Miwako Suzuki Yamanaka.

**Resources:** Miwako Suzuki Yamanaka.

**Supervision:** Yuri Hosokawa, Mamoru Ayusawa, Norikazu Hirose, Koji Kaneoka.

**Validation:** Miwako Suzuki Yamanaka, Yuri Hosokawa.

**Visualization:** Miwako Suzuki Yamanaka.

**Writing – original draft:** Miwako Suzuki Yamanaka.

**Writing – review & editing:** Miwako Suzuki Yamanaka, Yuri Hosokawa, Mamoru Ayusawa, Norikazu Hirose, Koji Kaneoka.

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
