## [Decision Letter · Decision Letter 0]

1 Jul 2021

PONE-D-21-19054

Epidemiology of Sports-Related Fatalities during Organized School Sports in Japanese High Schools between 2009 and 2018

PLOS ONE

Dear Dr. Yamanaka,

Thank you for submitting your manuscript to PLOS ONE. After careful consideration, we feel that it has merit but does not fully meet PLOS ONE’s publication criteria as it currently stands. Therefore, we invite you to submit a revised version of the manuscript that addresses the points raised during the review process.

Consider all comments of all reviewers including reviewer 3.

Please submit your revised manuscript by 15 July 2021. If you will need more time than this to complete your revisions, please reply to this message or contact the journal office at plosone@plos.org. Please include the following items when submitting your revised manuscript:

We look forward to receiving your revised manuscript.

Kind regards,

Ahmed Mancy Mosa, Ph.D.

Academic Editor

PLOS ONE

Journal Requirements:

Reviewers' comments:

Reviewer's Responses to Questions

**Comments to the Author**

1. Is the manuscript technically sound, and do the data support the conclusions?

Reviewer #1: Yes

Reviewer #2: Yes

Reviewer #3: Partly

2. Has the statistical analysis been performed appropriately and rigorously? 

Reviewer #1: Yes

Reviewer #2: N/A

Reviewer #3: No

3. Have the authors made all data underlying the findings in their manuscript fully available?

Reviewer #1: Yes

Reviewer #2: Yes

Reviewer #3: Yes

4. Is the manuscript presented in an intelligible fashion and written in standard English?

Reviewer #1: Yes

Reviewer #2: No

Reviewer #3: Yes

5. Review Comments to the Author

Reviewer #1: Original article. Simple and correct statistical study. I think the accident database is very useful and interesting. I think the total number is too low to make a correlation between the number of accidents and the type of sport practiced. the data should be correlated more with the frequency of sport. overall very interesting.

Reviewer #2: It is a pretty good summary of sport-related fatality in Japan. The amount of this work is enough and its analyses are comprehensive which can offer quite objective data for related stakeholder.

There are also some problems in this article: 1. language problem. For example, we say 'athletic trainer' instead of 'school nurse' professionally 2.analyses problem. This article doesn't mention the specific data of medical staff which I think is valuable for continuous improvement for school.

This is my suggestion above. Thank you very much.

Reviewer #3: In the manuscript by Suzuki-Yamanaka M. et al., the authors examined the frequency and incidence rate of sports-related fatalities in Japanese high schools between 2009 and 2018. Although an interesting and potentially important study, several issue should be considered.

1. The data was collected from insurance company and may be subjected to potential bias. Information on medical record or cross-validation with local or national death registry is lacking.

2. In the “method” part, description on data analysis is rather simple. It is unclear how the incidence rate is calculated and adjusted by what types of variables.

3. Conclusions are not supported by statistical analysis.

6. PLOS authors have the option to publish the peer review history of their article (what does this mean?). If published, this will include your full peer review and any attached files.

Reviewer #1: **Yes: **Giuseppe Attisani

Reviewer #2: No

Reviewer #3: No

---

## [Author Response · Author response to Decision Letter 0]

9 Jul 2021

Reviewer #1: Original article. Simple and correct statistical study. I think the accident database is very useful and interesting. I think the total number is too low to make a correlation between the number of accidents and the type of sport practiced. the data should be correlated more with the frequency of sport. overall very interesting.

Thank you very much for your valuable feedback on our work. 

We would like to note that in our original submission, we limited our calculation of the incidence rate of fatality in sports that reported five or more cases during the study period. Nevertheless, we acknowledge the concern raised by Reviewer 1, and have added the limitation regarding the small case number in the revised document (L339-341). 

Reviewer #2: It is a pretty good summary of sport-related fatality in Japan. The amount of this work is enough and its analyses are comprehensive which can offer quite objective data for related stakeholder.

There are also some problems in this article: 1. language problem. For example, we say 'athletic trainer' instead of 'school nurse' professionally 2.analyses problem. This article doesn't mention the specific data of medical staff which I think is valuable for continuous improvement for school.

This is my suggestion above. Thank you very much.

Thank you very much for your valuable feedback on our work. 

Thank you for pointing out the discrepancy regarding the nomenclature of profession. In Japan, most schools do not employ athletic trainers and they have school nurses to manage health and safety of student athletes. The use of school nurse as a term to describe the first responder was not by mistake but accurately reflects the situation in Japan. 

Regarding the medical staff data, we were unable to obtain information about the presence of medically trained personnel in most case reports submitted to the JSC database. Therefore, our finding is limited to what is already described in the original document. 

Reviewer #3: In the manuscript by Suzuki-Yamanaka M. et al., the authors examined the frequency and incidence rate of sports-related fatalities in Japanese high schools between 2009 and 2018. Although an interesting and potentially important study, several issue should be considered.

Thank you very much for your valuable feedback on our work. We have made changes to reflect your comments in the manuscript.

1. The data was collected from insurance company and may be subjected to potential bias. Information on medical record or cross-validation with local or national death registry is lacking.

We recognize the limitation of incidence rates calculation based on insurance claims. However, there currently is no additional mechanism to capture sports-related catastrophic incidents data in Japan. As almost all high schools and their students (97.7% in 2018) subscribe to the JSC Injury and Accident Mutual Aid Benefit System, and no competing companies exist for similar coverage, we believe the current data presented in our manuscript is the most reliable estimates.

2. In the “method” part, description on data analysis is rather simple. It is unclear how the incidence rate is calculated and adjusted by what types of variables.

Thank you for this comment. We have added details on how we carried incidence rate calculations for the subdivisions (year, sports, and type of fatality).

3. Conclusions are not supported by statistical analysis.

We modified the last sentence in the conclusion section to “To ensure the health and safety of student-athletes, distribution of medically trained personnel in high school sports and improved coaching education to appropriately respond to medical emergency should be promoted in Japan” given the finding that medically trained personnel were involved in initial care only in two cases, and the in the majority of cases, coaches were the first responder.

---

## [Decision Letter · Decision Letter 1]

26 Jul 2021

PONE-D-21-19054R1

Epidemiology of Sports-Related Fatalities during Organized School Sports in Japanese High Schools between 2009 and 2018

PLOS ONE

Dear Dr. Yamanaka,

Thank you for submitting your manuscript to PLOS ONE. After careful consideration, we feel that it has merit but does not fully meet PLOS ONE’s publication criteria as it currently stands. Therefore, we invite you to submit a revised version of the manuscript that addresses the points raised during the review process.

Please, carefully consider the comments

We look forward to receiving your revised manuscript.

Kind regards,

Ahmed Mancy Mosa, Ph.D.

Academic Editor

PLOS ONE

Journal Requirements:

Reviewers' comments:

Reviewer's Responses to Questions

**Comments to the Author**

1. If the authors have adequately addressed your comments raised in a previous round of review and you feel that this manuscript is now acceptable for publication, you may indicate that here to bypass the “Comments to the Author” section, enter your conflict of interest statement in the “Confidential to Editor” section, and submit your "Accept" recommendation.

Reviewer #3: All comments have been addressed

2. Is the manuscript technically sound, and do the data support the conclusions?

Reviewer #3: Yes

3. Has the statistical analysis been performed appropriately and rigorously? 

Reviewer #3: Yes

4. Have the authors made all data underlying the findings in their manuscript fully available?

Reviewer #3: Yes

5. Is the manuscript presented in an intelligible fashion and written in standard English?

Reviewer #3: Yes

6. Review Comments to the Author

Reviewer #3: The mansuscript has been modified to incorporate most of my suggestions. It seems that the "Conclusion" of the manuscript could still be specified and refined. Currently, the conclusion is a compilation of numbers and "jump" to a wide statement (the last sentence). The readers may benefit from more detailed conclusions like "fatality by indirect causes such cardiac failure dropped greatly due to XX policy" or "The high fatality rate in Judo was mainly caused by direct causes such as collisons that could be reduced by implement XX policy", rather than simply list the numbers.

7. PLOS authors have the option to publish the peer review history of their article (what does this mean?). If published, this will include your full peer review and any attached files.

Reviewer #3: **Yes: **Lei Quan

---

## [Author Response · Author response to Decision Letter 1]

2 Aug 2021

Reviewer #3: The manuscript has been modified to incorporate most of my suggestions. It seems that the "Conclusion" of the manuscript could still be specified and refined. Currently, the conclusion is a compilation of numbers and "jump" to a wide statement (the last sentence). The readers may benefit from more detailed conclusions like "fatality by indirect causes such cardiac failure dropped greatly due to XX policy" or "The high fatality rate in Judo was mainly caused by direct causes such as collisons that could be reduced by implement XX policy", rather than simply list the numbers.

Thank you very much for providing valuable feedback. We incorporated a couple of sentences in the conclusion section to better explain the findings of this study.

---

## [Decision Letter · Decision Letter 2]

6 Aug 2021

Epidemiology of Sports-Related Fatalities during Organized School Sports in Japanese High Schools between 2009 and 2018

PONE-D-21-19054R2

Dear Dr. Yamanaka,

We’re pleased to inform you that your manuscript has been judged scientifically suitable for publication and will be formally accepted for publication once it meets all outstanding technical requirements.

Kind regards,

Ahmed Mancy Mosa, Ph.D.

Academic Editor

PLOS ONE

Additional Editor Comments (optional):

Reviewers' comments:

Reviewer's Responses to Questions

**Comments to the Author**

1. If the authors have adequately addressed your comments raised in a previous round of review and you feel that this manuscript is now acceptable for publication, you may indicate that here to bypass the “Comments to the Author” section, enter your conflict of interest statement in the “Confidential to Editor” section, and submit your "Accept" recommendation.

Reviewer #3: All comments have been addressed

2. Is the manuscript technically sound, and do the data support the conclusions?

Reviewer #3: Yes

3. Has the statistical analysis been performed appropriately and rigorously? 

Reviewer #3: Yes

4. Have the authors made all data underlying the findings in their manuscript fully available?

Reviewer #3: Yes

5. Is the manuscript presented in an intelligible fashion and written in standard English?

Reviewer #3: Yes

6. Review Comments to the Author

Reviewer #3: In the manuscript by Yamanaka et al., the authors examined the frequency and incidence rate of sports-related fatalities in Japanese high schools between 2009 and 2018. After modfication, all my comments has been fulfilled.

7. PLOS authors have the option to publish the peer review history of their article (what does this mean?). If published, this will include your full peer review and any attached files.

Reviewer #3: **Yes: **Lei Quan

---

## [Editor Report · Acceptance letter]

12 Aug 2021

PONE-D-21-19054R2 

Epidemiology of Sports-Related Fatalities during Organized School Sports in Japanese High Schools between 2009 and 2018 

Dear Dr. Yamanaka:

I'm pleased to inform you that your manuscript has been deemed suitable for publication in PLOS ONE. Congratulations! Your manuscript is now with our production department. 

Kind regards, 

on behalf of

Dr. Ahmed Mancy Mosa 

Academic Editor

PLOS ONE